# Factors Predicting Response to Selective Retina Therapy in Patients with Chronic Central Serous Chorioretinopathy

**DOI:** 10.3390/jcm11020323

**Published:** 2022-01-10

**Authors:** Minhee Kim, Seung Hee Jeon, Ji-young Lee, Seung-hoon Lee, Young-jung Roh

**Affiliations:** 1Eye Hospital, Yeouido St. Mary’s Hospital, Catholic University of Korea, 10, 63-ro, Yeongdeungpo-gu, Seoul 07345, Korea; chriszz@naver.com (M.K.); jiyounglee.md@daum.net (J.-y.L.); 1124mynamelsh@gmail.com (S.-h.L.); 2Department of Ophthalmology and Visual Science, Incheon St. Mary’s Hospital, Catholic University of Korea, 56, Dongsu-ro, Bupyeong-gu, Incheon 21431, Korea; jsh881107@hanmail.net

**Keywords:** chronic central serous chorioretinopathy, real-time feedback-controlled dosimetry, selective retina therapy, subretinal fluid height

## Abstract

This retrospective study aimed to assess the safety and efficacy of selective retina therapy (SRT) with real-time feedback-controlled dosimetry (RFD) for chronic central serous chorioretinopathy (CSC) and to evaluate factors predictive of treatment response. We included 137 eyes of 135 patients with chronic CSC. SRT was performed to cover each of the leakage areas on fundus fluorescein angiography. Changes in mean best-corrected visual acuity (BCVA), central macular thickness (CMT), and subretinal fluid (SRF) height were evaluated at baseline and at 3 and 6 months after treatment. Complete SRF resolution was observed in 52.6% (72/137 eyes) and 90.5% (124/137 eyes) at 3 and 6 months, respectively. Mean BCVA (logMAR) significantly improved from 0.41 ± 0.31 at baseline to 0.33 ± 0.31 at month 6 (*p* < 0.001). Mean CMT significantly decreased from 347.67 ± 97.38 μm at baseline to 173.42 ± 30.95 μm at month 6 (*p* < 0.001). Mean SRF height significantly decreased from 187.85 ± 97.56 µm at baseline to 8.60 ± 31.29 µm after 6 months (*p* < 0.001). Baseline SRF height was a significant predictive factor for retreatment requirement (*p* = 0.008). In conclusion, SRT showed favorable anatomical outcomes in patients with chronic CSC. A higher baseline SRF height was a risk factor for retreatment.

## 1. Introduction

Central serous chorioretinopathy (CSC) is characterized by idiopathic neurosensory retinal detachment at the macula with or without retinal pigment epithelium (RPE) alterations [1,2]. Its incidence is approximately 1 per 10,000 cases and is highest in middle-aged men [3]. Although the pathogenesis of CSC is unclear, extensive hyperpermeable choroidal circulation is thought to be associated with subretinal fluid (SRF) accumulation and pigment epithelial detachment (PED) [4,5]. Since RPE dysfunction causes the breakdown of the outer blood-retinal barrier, it may also cause SRF and PED [6]. In acute CSC, SRF resolves spontaneously within 1–4 months, with good visual recovery [7], whereas in chronic CSC, permanent vision loss occurs because of diffuse RPE atrophy, subretinal fibrosis, and posterior cystoid retinal degeneration in the areas of non-resolving SRF [1,8]. Therefore, the goal of treatment is to eliminate SRF and prevent further visual impairment in CSC patients with long-lasting SRF [9,10].

Although there is no standard treatment for CSC, various treatment methods have been used, including conventional laser photocoagulation, photodynamic therapy (PDT), intravitreal anti-vascular endothelial growth factor (anti-VEGF) injection, and mineralocorticoid receptor antagonists, with favorable outcomes in several clinical studies. However, all treatments have been associated with ocular and systemic adverse events [1]. For instance, conventional laser photocoagulation causes central scotoma and choroidal neovascularization due to irreversible retinal damage [11,12]. Acute vision loss due to RPE atrophy and subretinal or sub-RPE hemorrhage has been reported after PDT with full-dose or reduced-dose settings [13,14]. While off-label use of intravitreal anti-VEGF agents has been effective for CSC [15,16], the association between cardiovascular risks and intravitreal anti-VEGF therapy remains controversial [17]. In addition, non-damaging retinal lasers, such as subthreshold micropulse laser and endpoint management, showed favorable outcomes in several studies [18,19,20]. However, the efficacy of subthreshold lasers for CSC should be confirmed in a randomized controlled study.

Since the adverse effects of conventional laser photocoagulation hamper its use in macular diseases, selective retina therapy (SRT) delivering microsecond pulses (1.7 µs) has been developed to produce selective RPE damage while sparing the photoreceptors and Bruch’s membrane [21]. Selective RPE damage can be achieved by using multiple short-duration micropulses and manipulating the pulse frequency [22,23]. Post-SRT RPE-damaged lesions were repaired within 7 days by migration and proliferation of adjacent RPE cells [24,25]. Although the SRT mechanism is unclear, restoration of a new RPE layer in the SRT-treated area might play a role in RPE rejuvenation [21]. Additionally, the release of cell mediators, such as matrix metalloproteinase-2 (MMP-2) and pigment epithelium-derived factor (PEDF), during the process of RPE restoration is beneficial [26,27]. Considering that no photoreceptor loss has been observed during RPE restoration post-SRT, the use of SRT for treating macular diseases with RPE abnormalities, such as age-related macular degeneration (AMD) and CSC, has been investigated [28,29,30,31,32,33,34,35].

Unlike conventional lasers, which generate visible lesions at laser spots, SRT spots are invisible during irradiation because SRT lesions are confined to the RPE layer without damaging the adjacent neurosensory retina. To confirm appropriate SRT lesions, two clinical endpoints, including an “invisible spot on ophthalmoscopy” and a “visible spot on fundus fluorescence angiography (FFA)” have been used in all clinical studies [28,29,30,31,32,33,34,35]. As SRT micropulses disrupt RPE cells by producing short-lived microbubbles in the melanosome, optoacoustic dosimetry and reflectometry have been developed for real-time monitoring of microbubbles as RPE damage indicators. Recently, real-time feedback-controlled dosimetry (RFD) using both optoacoustic dosimetry and reflectometry has been optimized [25,33,34,35,36,37,38,39]. Briefly, when microbubbles occur after SRT, optoacoustic dosimetry monitors ultrasonic pressure wave signals [21,22,23], whereas reflectometry monitors the modulation of backscattered light signals [37,38,40]. Although physicians can adjust the preset pulse energy of each irradiation on the basis of RFD feedback signals, to avoid over- or under-treatment, pre-treatment FFA remains useful for obtaining the overall safety margin of the pulse energy.

This study sought to investigate the long-term efficacy and safety of SRT using RFD and to identify predictors of the response to SRT in a large cohort of patients with chronic CSC.

## 2. Materials and Methods

We retrospectively reviewed the medical records of CSC patients who underwent SRT with RFD between September 2016 and February 2019. Potential risks and benefits related to SRT were discussed with all patients, who gave written informed consent before SRT treatment. The inclusion and exclusion criteria are presented in Table 1.

### 2.1. SRT Procedure

One retina specialist (YJR) performed SRT (Q-switched Nd:YLF 527-nm laser, 1.7-μs micropulse duration) using an SRT device equipped with RFD (R:GEN, Lutronic, Goyang-si, South Korea), approved by the Ministry of Food and Drug Safety in South Korea for CSC. A specified contact lens with an embedded ultrasonic transducer (Lutronic) was used to deliver a 200 μm diameter spot onto the retina. According to the implemented SRT laser system settings, a maximum of 15 micropulses per burst were delivered with escalating energies. The first micropulse’s energy was 50% of that of the 15th micropulse. The energy increased by 3.57% per micropulse. One hour after irradiating the test spots around the arcade vessels, FFA was performed to determine the appropriate energy for treatment spots. The minimum pulse energies indicating FFA-positive SRT spots were chosen for the initial irradiation of the treatment spots.

SRT was applied to cover the entire leakage area on FFA with a one-spot spacing density (Figure 1).

The PED was directly irradiated with the SRT if it was related to a leak. Because SRT spots were invisible during irradiation, the one-spot spacing density between the spots was maintained using a guide-beam.

The physician controlled the preset pulse energy by adjusting the 15th pulse energy as follows. The RFD threshold was set to 2.0 arbitrary units (AU) for optoacoustic dosimetry and 6.0 AU for reflectometry. If one of the micropulses of each irradiation reached either an optoacoustic value >2.0 AU or reflectometric value >6.0AU, subsequent micropulse irradiation stopped automatically (so-called auto-stop) (Figure 2). The correlation between RFD value and FFA-positive spot was evaluated on the basis of angiographic features of test spots because leaking points of CSC hinder the interpretation of angiographic features of treatment spots.

On the basis of the auto-stop occurring between the 1st and 15th micropulses, the RFD displayed an upward-pointing arrow (undertreatment alarm), a sideways-pointing arrow (appropriate treatment), or a downward-pointing arrow (overtreatment alarm). Accordingly, the preset treatment spot energy was increased or decreased instantly by a 10-µJ or 20-µJ step, as previously described [34].

If the SRF height increased or persisted on OCT 3 months after the initial treatment, SRT was repeated with the same initial preset pulse energy used in the initial SRT. When residual SRF was observed, retreatment was also performed. A similar number of treatment spots were irradiated to cover the same area of leakage on the FFA. However, if the SRF almost resolved (SRF height <10 μm), no additional SRT was performed. Recurrence was defined as the development of new SRF on OCT images after complete SRF resolution after the initial SRT. Adverse effects of SRT were also documented.

### 2.2. Clinical Measures

A complete ophthalmological examination, including slit-lamp evaluation and best-corrected visual acuity (BCVA), was performed at baseline and 3 and 6 months post-SRT. BCVA was measured using a standard Snellen chart and was converted to the logarithm of the minimum angle of resolution (logMAR) for analysis. Color fundus photographs (CFP) (CF-60UVi, Canon Inc., Tokyo, Japan) and fundus autofluorescence (FAF) (HRA2; Heidelberg Engineering, Dossenheim, Germany) images were taken at each visit.

FFA (HRA2) or ultra-wide-field FFA (Optos P200TDX, Optos PLC, Dunfermline, Scotland, UK) were performed in all patients at baseline and on the treatment day. FFA was repeated for retreatment if the SRF was sustained or increased during follow-up. On the basis of the initial FFA, patients were categorized as having either focal or diffuse leakage. Focal type was defined as a maximum of three pinpoint leakages on FFA, and diffuse type was defined as more than three leakages or areas of diffuse hyperfluorescent leakages on FFA.

Spectral-domain OCT (Cirrus, Carl Zeiss Meditec, Dublin, CA, USA) was employed to detect SRF and to measure the central macular thickness (CMT) and the SRF height using the macular cube 512 × 128 scan protocol in a central 6 × 6 mm^2^ area at each visit. The SRF height at the foveal center was measured as the distance between the outer neurosensory retina and the RPE. At the leakage site on FFA images, SRF and PED heights were measured based on OCT at the initial visit. PED was categorized as “present” if PED showed a dome-shaped or flat irregular type and as “absent” if there were no PED or RPE bumps.

### 2.3. Statistical Analysis

Changes in logMAR BCVA, CMT, and SRF height from baseline to months 3 and 6 after the initial SRT were analyzed using repeated-measures analysis of variance with Greenhouse–Geisser correction. Post hoc tests using Bonferroni correction were performed to assess changes between baseline and each follow-up visit. Comparison between complete SRF resolution and remnant SRF groups at 3 months was performed using Student’s *t*-test for nominal variables (age, symptom duration, baseline BCVA, baseline CMT, and baseline SRF height) and chi-square test for categorical variables (sex, type of leakage, PED type, and history of anti-VEGF injection).

Patients with complete SRF resolution at 6 months were divided into the single-SRT group, which underwent a single SRT session, and the retreatment group, which underwent additional SRT. In patients with complete SRF resolution at month 6, variables were compared between the single-SRT group and retreatment group using Student’s *t*-test and the chi-square test. Multiple logistic regression analyses were performed to determine factors associated with retreatment. A receiver operating characteristic (ROC) curve was plotted using the baseline SRF height for predicting retreatment.

Statistical analyses were performed using SPSS (v24.0; SPSS Inc., Chicago, IL, USA) and R Statistical Software v3.6.2 (R Foundation for Statistical Computing, Vienna, Austria). Statistical significance was set at *p* < 0.05.

## 3. Results

Among the 191 eyes (187 patients) who underwent SRT for chronic CSC over the study period, 137 eyes (135 patients) met the inclusion criteria. Twenty-one eyes were excluded because the follow-up period was <6 months. Fifteen eyes with a history of intravitreal bevacizumab injections <12 weeks pre-SRT were excluded. Eleven eyes with a history of PDT and five eyes with a history of conventional laser therapy were excluded. Two eyes with pathological myopia were excluded from the study.

The demographic and clinical features of the patients are summarized in Table 2. At baseline, the mean age of the patients was 48.2 ± 8.8 years. There were 82.2% male patients (111 eyes) and 17.8% female patients (24 eyes).

Changes in BCVA, CMT, and SRF height during the 6-month follow-up are summarized in Table 3. Mean BCVA changes from baseline to each follow-up visit (month 3, month 6) were significant (F (1.522, 206.976) = 15.628, *p* < 0.001). Post hoc testing using Bonferroni correction revealed statistically significant improvement in logMAR BCVA from baseline (0.41 ± 0.31) to month 3 (0.34 ± 0.32, *p* = 0.001) and to month 6 (0.33 ± 0.31, *p* < 0.001). However, BCVA was similar at months 3 and 6 (*p* > 0.99).

Mean CMT changed significantly from baseline to each follow-up visit (month 3, month 6) (F (1.904, 258.948) = 225.447, *p* < 0.001). There was a significant improvement in mean CMT from baseline (347.67 ± 97.38 μm) to month 3 (222.23 ± 85.34 μm, *p* < 0.001) and to month 6 (173.42 ± 30.95 μm, *p* < 0.001). CMT differed significantly between months 3 and 6 (*p* < 0.001).

Mean SRF height changed significantly from baseline to each follow-up visit (F (2, 272) = 235.419, *p* < 0.001). The mean SRF height at baseline (187.85 ± 97.56 μm) decreased significantly to month 3 (62.41 ± 85.41 μm, *p* < 0.001) and to month 6 (8.6 ± 31.29 μm, *p* < 0.001). Additionally, there was a significant difference between months 3 and 6 (*p* < 0.001).

At 3 months post-SRT, 72 eyes showed complete SRF resolution (SRF resolution group), whereas 65 eyes showed remnant SRF (remnant SRF group). A comparison of baseline characteristics between the complete SRF resolution and remnant SRF groups is shown in Table 4. The type of leakage (*p* = 0.046), mean CMT (*p* = 0.027), and SRF height (*p* = 0.009) differed significantly between the groups.

Complete SRF resolution on OCT was observed in 72 eyes (52.6%) at month 3 and in 124 eyes (90.5%) at month 6. SRF persisted in 13 eyes at 6 months post-SRT. Among the thirteen eyes, six and five eyes showed a decrease of ≥50% and <50% in SRF height, respectively, compared to baseline SRF. However, two eyes showed increased SRF during the 6-month follow-up period. While 48 eyes required retreatment at month 3 due to persistent SRF, 17 eyes did not. In 48 retreated eyes, the mean SRF height decreased significantly from month 3 (144.67 ± 83.91 μm) to month 6 (12.46 ± 32.14 μm, *p* < 0.001). Among the 17 eyes, 15 showed complete SRF resolution at 6 months. Additionally, among the 72 eyes that showed complete SRF resolution at month 3, SRF recurred in three eyes at 6 months post-treatment. Ten eyes with remnant SRF did not respond to SRT at 6 months post-treatment (Figure 3). Among the 10 eyes, 7 eyes showed diffuse leakages. In addition, seven eyes showed flat irregular PED, whereas the other three eyes had RPE bumps.

Overall, among 124 eyes with complete SRF resolution at 6 months after SRT, 84 eyes received a single session of SRT (single SRT group), whereas 40 eyes needed additional SRT (retreatment group). Comparisons of baseline characteristics between the single SRT and retreatment groups are shown in Table 5.

Only CMT (*p* = 0.004) and SRF height (*p* = 0.009) differed significantly between the groups. Multiple logistic regression analysis revealed that baseline SRF height predicted retreatment need. Eyes with higher SRF height at baseline had a higher odds ratio for retreatment (*p* = 0.008; Table 6). The ROC curve analysis revealed that the optimal cutoff value (Youden index) was an SRF height of 133.5 μm (sensitivity 82.5%, specificity 44%, area under the receiver operating characteristic curve 0.661, *p* = 0.004) (Figure 4). Retreatment was performed in 41.3% (33/80 eyes) of eyes with an SRF height ≥133.5 μm and in 15.9% (7/44 eyes) of eyes with SRF height <133. 5 μm.

In both the initial and retreatment SRT, the preset pulse energy ranged between 60 and 180 μJ. The mean preset pulse energy for the initial and retreatment SRT were 131.3 ± 38.2 μJ and 138.5 ± 37.5 μJ, respectively. The mean actually applied pulse energies for the initial and retreatment SRT determined by RFD were 98.3 ± 32.5 μJ (range 65.0–137.5 μJ) and 103.3 ± 33.2 μJ (range 66.5–135.0 μJ). The mean numbers of SRT spots for the initial SRT and retreatment were 23.9 ± 15.2 and 22.3 ± 12.4, respectively. The ratio of automatic-stopped spots to FFA-positive test spots was 93.9% (1930/2055 spots).

All SRT spots applied in this study were invisible on the CFP during the 6-month follow-up. Dark dots on FAF indicating RPE atrophy were not observed in SRT spots over the follow-up period. Additionally, no SRT-related adverse events, such as retinal hemorrhage or burns, were observed in this study.

## 4. Discussion

To our knowledge, no previous study included such a large number of chronic CSC patients treated with SRT. We found that SRT with RFD showed favorable anatomical outcomes in chronic CSC patients, and we identified a greater baseline SRF height as a predictor of retreatment requirement.

Previous studies with different patient numbers, inclusion criteria, and follow-up periods found a complete SRF resolution rate at 3 months post-SRT of 65–75% in chronic CSC patients [30,32,33,34]. Although complete SRF resolution was achieved in 52.6% (72/137) of eyes in the present study, it improved to 90.5% (124/137) at 6 months post-SRT. This improved SRF resolution rate was not only because 83.3% (40/48) of retreated eyes showed complete SRF resolution, but also because 15 of 17 (88.2%) eyes with minimal SRF at month 3 showed complete SRF resolution at 6 months post-SRT. Since 17 eyes with a residual SRF height of <10 μm almost showed SRF resolution after the initial SRT, observation was preferred to retreatment. The complete resolution rate of 88.2% (15 of 17 eyes) at 6 months might be due to the first SRT rather than spontaneous resolution. While BCVA, CMT, and SRF height improved statistically significantly between visits, BCVA remained similar between 3 and 6 months. However, the logMAR BCVA improved continuously from month 3 to month 6.

Comparative analysis between complete SRF resolution and remnant SRF groups at 3 months post-SRT showed that the proportion of diffuse leakage, mean CMT, and mean SRF height were significantly higher in the remnant SRF group. Therefore, a large SRF load and multiple leakages may adversely influence the effect of SRT. 

Subgroup analysis was performed in the complete SRF resolution group to evaluate predictors of retreatment, and only initial SRF height was identified as significant by multivariate analysis. Since a higher SRF height represents a larger SRF volume, CSC patients with large SRF volumes may need additional SRT, regardless of other factors. The optimal baseline SRF height cutoff value for retreatment was 133.5 μm. In the single-SRT group, 47/84 eyes (56%) had an SRF height ≥133.5 μm, compared to 33/40 eyes (83%) in the retreatment group. Therefore, a greater SRF height increased the retreatment risk. We suspect that the potency of the new RPE layer for removing SRF is confined to the SRT-treated area, and it would require more time to resolve a large SRF volume.

In CSC, a single point leakage on FFA is regarded as focal leakage, whereas multiple point leakages and ill-defined leakage can be regarded as diffuse leakage [41]. The leakage area is presumed to be associated with a small tear RPE or focal outer blood-retinal barrier defect [10]. The new RPE layer induced by SRT might have gained improved integrity of the outer blood-retinal barrier, which could lead to improved function of RPE cells. Apart from the improvement of RPE function, the closure of leaking points by new RPE cells might be also helpful to remove SRF. In addition, the release of cell mediators, such as PEDF and MMP-2 in SRT-treated areas were observed during RPE restoration [27]. These cell mediators have been considered mainly as the related factors in the treatment of AMD because PEDF is associated with the inhibition of vascular permeability by regulating the angiogenic effect of VEGF [42,43], and MMP-2 is related to the improvement of the flux across the Bruch’s membrane [44]. However, considering that anti-VEGF agents show some therapeutic effect in patients with CSC, anti-angiogenic PEDF might play a role in SRF resolution. Considering that the cause of CSC is known to be associated with choroidal hyperpermeability or RPE dysfunction [4,5,6], SRT might mainly improve RPE function and outer blood-retinal barrier by forming a new RPE layer. Therefore, retreatment of SRT could be helpful to expand the area of the new RPE layer, which could provide better clinical outcomes for predicted poor responders, such as patients with a large SRF volume.

Among the 13 eyes with persistent SRF 6 months post-SRT, three eyes with complete SRF resolution at 3 months post-SRT had recurrence at 6 months post-SRT. As most previous studies used 3-month follow-up periods, the post-SRT recurrence rate has not been well-investigated. Considering that the long-term recurrence rate of untreated CSC was previously reported to be approximately 52% [45], the 4% (3/72 eyes) post-SRT recurrence rate seems to be low, although our follow-up period was short. 

Although RFD was used to titrate the preset pulse energy during irradiation, the initial preset pulse energy of the treatment spots was chosen according to the angiographic features of the test spots around the arcade vessels. Because the safety range of the pulse energy of the test spot was confirmed before irradiating the treatment spots, pulse energy adjustments could be controlled within the safety margin during irradiation in the macular area. Regarding test spot evaluation, RFD detected 93.9% of FFA-positive spots in this study. The 5.9% of FFA-positive spots that RFD missed from the detection showed very faint leakages which indicate weak RPE damage. The remaining FFA-positive spots (0.2%) were missed because of the misalignment of contact lens or pedal error. Although RFD-guided SRT did not induce overtreatment in the current study, pretreatment FFA is still necessary to avoid overtreatment.

The efficacy of PDT and subthreshold micropulse laser (SMPL) was previously compared in a randomized trial [46]. In the PLACE trial, the complete SRF resolution rate (67%) of the half-dose PDT group was superior to that of the SMPL group (29%) 6–7 months post-treatment. Although SRT is considered a type of SMPL because of common characteristics, such as “invisibility during irradiation” and “photoreceptor-sparing,” the laser pulse length used in SRT is much shorter (1.7 µs) than that used in SMPL (100–300 µs). In addition, tissue temperature rise in SMPL is mainly determined by the duty cycle, which is the frequency of the train of micropulses. The therapeutic effect of SMPL is associated with simple retinal photostimulation, such as RPE heat shock protein activation without any retinal tissue damage [29]. SMPL can cause the temperature rise of RPE cells if the rise is high enough, damaging RPE cells thermally, whereas SRT destroys RPE cells mechanically through microbubble formation in the melanosomes of the RPE cells. Therefore, our results should be differentiated from those of other SMPLs because of the differences in SRT and SMPL.

This study has several limitations. First, a retrospective and nonrandomized study design was used, and the follow-up period was relatively short. Second, although CSC is known to be a pachychoroid disease, choroidal thickness was not investigated because the data were not available for all patients. In a previous study, we demonstrated that the effect of SRT was primarily restricted to the RPE, rather than the choroid, because the change in choroidal thickness post-SRT was insignificant [34]. However, a recent study reported that a decrease in choroidal thickness post-SRT was significant [47]. The difference in the change in choroidal thickness might be due to the different demographics and patient numbers. Further prospective studies regarding choroidal thickness are needed to understand the mechanism of SRT. Third, risk factors for CSC, such as smoking history and alcohol consumption, were not investigated in this study.

## 5. Conclusions

In conclusion, our study demonstrated that SRT with RFD was effective in removing SRF during a 6-month follow-up period. A greater initial SRF height predicted retreatment requirement. Further large prospective studies are necessary to confirm the predictive factors affecting the response to SRT.

## Figures and Tables

**Figure 1 jcm-11-00323-f001:**
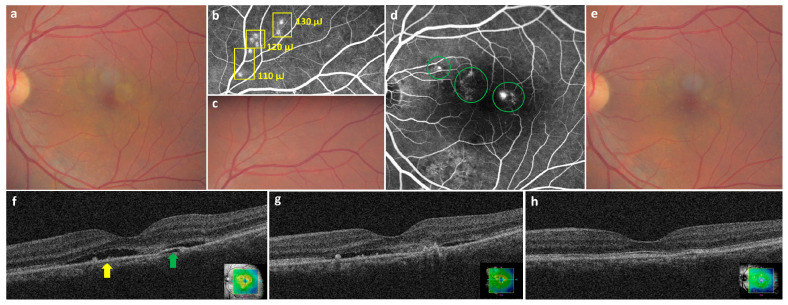
Representative pictures of the selective retinal therapy (SRT) procedure in the left eye of a 49-year-old man. He presented with a 12-months history of blurred vision in the left eye. His best-corrected visual acuity (BCVA) in the left eye was 20/32. (**a**) At baseline, subretinal fluid (SRF) was observed on color fundus photography (CFP). (**b**) Seven hyperfluorescent test spots (110–130 μJ) (yellow rectangle) were observed around the superior temporal vessels on fundus fluorescein angiography (FFA) 1 h after irradiation. (**c**) The 7 test spots were invisible on CFP 1 h after irradiation. (**d**) Thirty-one SRT spots were applied at multiple leaking points (green circles) on FFA. (**e**) No visible SRT spot was seen 6 months post-treatment (**f**) SRF (yellow arrow) with pigment epithelial detachment (green arrow) was observed on baseline optical coherence tomography (OCT). (**g**) Since SRF persisted on OCT images obtained 3 months after SRT, retreatment was performed. (**h**) SRF was completely resolved on OCT images obtained at 6 months post-treatment, and BCVA had improved to 20/25.

**Figure 2 jcm-11-00323-f002:**
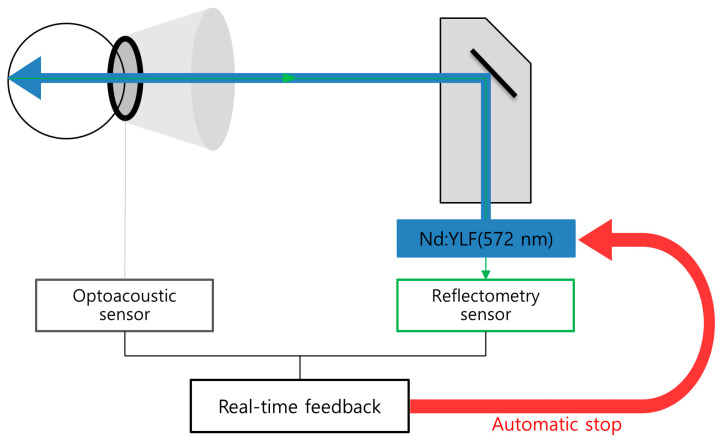
A schematic diagram representing the signal processing of real-time feedback-controlled dosimetry. Acoustic transient signals are detected by a contact lens with an inserted ring-shaped optoacoustic sensor and backscattered light signals are detected by a reflectometry sensor. When the acoustic or light feedback signals reach the set thresholds, irradiation stops automatically.

**Figure 3 jcm-11-00323-f003:**
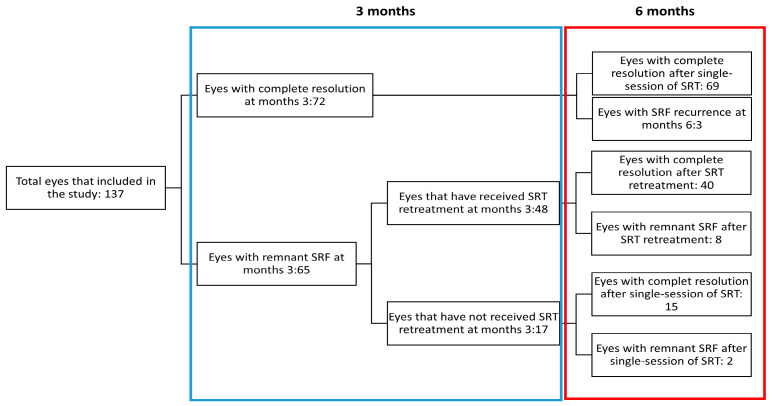
Flow-chart showing the rate of complete subretinal fluid resolution at 3 months and 6 months following selective retina therapy.

**Figure 4 jcm-11-00323-f004:**
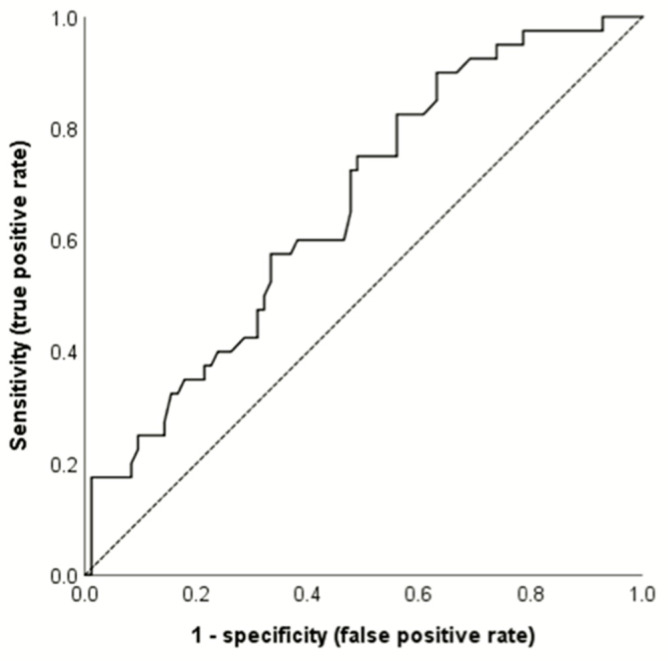
A receiver operating characteristic curve was calculated using the baseline subretinal fluid (SRF) height for predicting the need for retreatment. The optimal cutoff value (Youden index) was determined as an SRF height of 133.5 μm (sensitivity 82.5%, specificity 44%, area under the receiver operating characteristic curve 0.661, *p* = 0.004).

**Table 1 jcm-11-00323-t001:** Inclusion and exclusion criteria.

Inclusion Criteria
CSC patients who underwent SRT
Presence of SRF involving the fovea on OCT images for ≥3 months
Presence of focal or diffuse leakages on FFA caused by CSC
Availability of ≥6 months of medical records after the initial SRT
Age ≥ 18 years
**Exclusion Criteria**
Presence of other macular diseases, including AMD, polypoidal choroidal vasculopathy (PCV), and pathological myopia
Presence of an RPE atrophy area > 500 μm diameter
History of conventional laser photocoagulation or PDT for CSC
History of intravitreal bevacizumab injection ≤ 12 weeks prior to SRT
Media opacity that could interfere with SRT irradiation or adequate acquisition of FFA, indocyanine green angiography, FAF, and OCT images

CSC: central serous chorioretinopathy, SRT: selective retina therapy, SRF: subretinal fluid, OCT: ocular coherence tomography, FFA: fundus fluorescein angiography, AMD: age-related macular degeneration, RPE: retinal pigment epithelium, PDT: photodynamic therapy, FAF: fundus autofluorescence.

**Table 2 jcm-11-00323-t002:** Baseline demographics and clinical findings of total patients with chronic central serous chorioretinopathy (CSC).

Patients’ Characteristics	Values
Number of patients (eyes)	135 (137)
Age, years, mean ± SD (range)	48.2 ± 8.8 (29–69)
Gender, *n* (%)	Male 111 (82.2%)/Female 24 (17.8%)
Bilaterality, *n* (%)	2 (1.5%)
Symptom duration in months, mean ± SD (range)	15.8 ± 21.2 (3–120)
Previous treatments	
Patients who received intravitreal bevacizumab injection, *n* (%)	58 (43%)
Type of leakages	
Focal, *n* (%)	84 (61.3%)
Diffuse, *n* (%)	53 (38.7%)
Presence of PED	
no PED or RPE bumps, *n* (%)	44 (32.1%)
PED (dome, flat irregular), *n* (%)	93 (67.9%)
Baseline BCVA (LogMAR), mean ± SD (range)	0.41 ± 0.31 (0–1.0)
Baseline CMT, µm, mean ± SD (range)	347.67 ± 97.38 (228–808)
Baseline SRF height, µm, mean ± SD (range)	187.85 ± 97.56 (18–648)

SD: standard deviation, PED: pigment epithelium detachment, RPE: retinal pigment epithelium, BCVA: best corrected visual acuity, LogMAR: logarithm of the minimum angle of resolution, CMT: central macular thickness, SRF: subretinal fluid.

**Table 3 jcm-11-00323-t003:** Best-corrected visual acuity, central macular thickness, and subretinal fluid height change during follow-up of patients with chronic central serous chorioretinopathy treated with selective retina therapy with real-time feedback dosimetry.

	Baseline	3 M	6 M	*p* Value *	*p* Value **	*p* Value ***
Best-corrected visual acuity, logMAR, mean ± SD	0.41 ± 0.31	0.34 ± 0.32	0.33 ± 0.31	0.001	<0.001	>0.99
Central macular thickness, µm, mean ± SD	347.67 ± 97.38	222.23 ± 85.34	173.42 ± 30.95	<0.001	<0.001	<0.001
Subretinal fluid height, µm, mean ± SD	187.85 ± 97.56	62.41 ± 85.41	8.6 ± 31.29	<0.001	<0.001	<0.001

* comparison between baseline and 3M; ** compared between baseline and 6M; *** compared between 3 months and 6 months; post hoc test using Bonferroni correction.

**Table 4 jcm-11-00323-t004:** Comparison of baseline characteristics between complete SRF resolution and remnant SRF groups 3 months after selective retina therapy.

	SRF Resolution Group	Remnant SRF Group	*p* Value
Number of eyes	72	65	
Age, years, mean ± SD (range)	47.3 ± 9.4	49.8 ± 9.4	0.123
Gender, *n* (%)	Male 61 (84.7%)/Female 11 (15.3%)	Male 52 (80.0%)/Female 13 (20.0%)	0.468
Symptom duration in months, mean ± SD	15.7 ± 21.9	15.9 ± 20.4	0.948
Previous treatments			
Patients who received intravitreal bevacizumab injection, *n* (%)	24 (33.3%)	34 (52.3%)	0.343
Type of leakages			0.046
Focal, *n* (%)	50 (69.4%)	34 (52.3%)	
Diffuse, *n* (%)	22 (30.6%)	31 (47.7%)	
Presence of PED			0.748
no PED or RPE bumps, *n* (%)	24 (33.3%)	20 (30.8%)	
PED (dome, flat irregular), *n* (%)	48 (66.7%)	45 (69.2%)	
Baseline BCVA (LogMAR), mean ± SD	0.4 ± 0.3	0.5 ± 0.3	0.107
Baseline CMT, μm, mean ± SD	330.4 ± 99.6	366.8 ± 91.8	0.027
Baseline SRF height, μm, mean ± SD	170.4 ± 99.6	207.2 ± 92.2	0.009

SRF, subretinal fluid; SD, standard deviation; PED, pigment epithelium detachment; BCVA, best corrected visual acuity; LogMAR, logarithm of the minimum angle of resolution; CMT, central macular thickness.

**Table 5 jcm-11-00323-t005:** Comparison of baseline characteristics between single-selective retina therapy group and retreatment group with chronic central serous chorioretinopathy patients who showed complete subretinal fluid resolution.

	Single-SRT Group	Retreatment Group	*p* Value
Number of eyes	84	40	
Age, years, mean ± SD (range)	47.6 ± 9.7	49.1 ± 8.8	0.402
Gender, *n* (%)	Male 72 (85.7%)/Female 12 (14.3%)	Male 32 (80.0%)/Female 8 (20.0%)	0.584
Symptom duration in months, mean ± SD	15.6 ± 21.1	17.3 ± 23.9	0.685
Previous treatments			
Patients who received intravitreal bevacizumab injection, *n* (%)	32 (38.1%)	20 (50.0%)	0.224
Type of leakages			0.29
Focal, *n* (%)	56 (66.7%)	22 (55.0%)	
Diffuse, *n* (%)	28 (33.3%)	18 (45.0%)	
Presence of PED			0.266
no PED or RPE bumps, *n* (%)	31 (36.9%)	10 (25.0%)	
PED (dome, flat irregular), *n* (%)	53 (63.1%)	30 (75.0%)	
Baseline BCVA (LogMAR), mean ± SD	0.4 ± 0.3	0.5 ± 0.3	0.096
Baseline CMT, μm, mean ± SD	331.4 ± 94.3	378.9 ± 90.2	0.004
Baseline SRF height, μm, mean ± SD	171.5 ± 94.4	219.0 ± 90.2	0.009

SD: standard deviation, PED: pigment epithelium detachment, BCVA: best corrected visual acuity, LogMAR: logarithm of the minimum angle of resolution, CMT: central macular thickness, SRF: subretinal fluid.

**Table 6 jcm-11-00323-t006:** Multiple logistic regression analysis of factors associated with need for SRT retreatment in patients with chronic central serous chorioretinopathy.

	OR	95% CI	*p* Value
Baseline SRF height	1.006	1.002–1.011	0.008
Baseline BCVA	2.807	0.745–10.569	0.750
Age	1.001	0.956–1.048	0.961
Gender	1.845	0.620–5.494	0.271
Symptom duration	1	0.979–1.020	0.973
Previous history of intravitreal bevacizumab injection	0.988	0.861–1.134	0.867
Type of leakages	1.304	0.515–3.299	0.575
Presence of PED	1.619	0.620–4.227	0.326

OR, odds ratio; CI, confidence interval; SRF, subretinal fluid; BCVA, best-corrected visual acuity; PED, pigment epithelium detachment; SRT, selective retina therapy.

## Data Availability

Data available on request because of restrictions, e.g., privacy or ethical restrictions.

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
