# Peer review of "Factors Predicting Response to Selective Retina Therapy in Patients with Chronic Central Serous Chorioretinopathy"

_jcm, 2022, doi:10.3390/jcm11020323_

Round 1
Reviewer 1 Report
Factors predicting response to selective retina therapy in patients with chronic central serous chorioretinopathy by Kim et al.
Authors reported their retrospective study with a large number of patients of chronic CSC treated with SRT. Clinical results as well as the statistical analysis regarding the factor having a high association with the retreatment at 3 months. Results showed that the high initial SRF hight is significantly related to the retreatment at 3 months.
The excellent clinical efficacy of SRT for chronic CSC has been clearly reported. However, the report on the predictor, which is also the title of this article, is problematic in several ways: Authors describe the difference in the response only with respect to the necessity of the retreatment, but not with the clinical features. The readers can have only one information with respect to the difference among the groups at 3 months, whether or not the patients showed remained SRF.
it is not clear what differences there are in clinical characteristics between the group that was retreated because of residual SRF after 3 months (48 patients) and the group that still did not receive retreatment (17 patients). There is no criterion for retreatment other than "residual SRF" after 3 months. Was their SRF less than the baseline, unchanged or increased? How did the Ophthalmologist decide to do or not to do the retreatment? Without clarification of this point, the meaning of re-treatment becomes ambiguous. Therefore, it is required to show the clinical characteristics of the 65 patients who did not completely disappear in 3 months more in detaile.
It is easy to assume that the higher the SRF, the longer it will take for the fluid to disappear (independent of the treatment method). If the leakage point is closed at the first SRT and remains closed after 3 months, isn't it possible that it will disappear if waited a little longer? In fact, 15 out of 17 patients who did not undergo re-treatment showed spontaneous disappearance.
If authors would like to point that the second SRT is absolutely needed and re-activate or accelerate the fluid resolution, then authors are asked to show the difference in SRF changes between the first 3 months and the subsequent 3 months for these patients.
In discussion, authors write that only the area irradiated by the SRT laser could have a function to remove SRF, but it could be not really correct, though it is agreed that SRT stimulates that. RPE cells have as their basic function to resolve subretinal fluid. So, it is considered that the leakage point is closed by any laser treatment (photocoagulation, SRT, micropulse, PDT…), other healthy RPE cells can easily pump out the fluid.
Also, which cell mediators do authors indicate here (line 280)?
Once again, the criteria for retreatment needs to be more clear and it is necessary to provide clinical data to support the author's claim of “differences in clinical response”. If not, the potentially misleading title should be changed.
Minor point: In introduction (lines between 40-49), please add the minimally invasive thermal laser treatment, such as the one using a subthreshold microsecond pulsed laser (SML) or endpoint management. There are already a lot of reports showing their effects for CSC.
Reviewer 2 Report
The authors retrospectively assessed the safety and efficacy of SRT for chronic CSC and evaluated factors predictive of treatment response.
This paper was nicely written but the previous reports including their prior study had already evaluated the safety and efficacy of this treatment for CSC. A result that a higher baseline SRF height was a risk factor for retreatment is interesting. The authors should add more findings to make readers understand the usefulness of SRT.
Additionally, I have following concerns.
- Please indicate the characteristics of 10 eyes with remnant SRF at 6 months, including the number and the intensity of FFA leakages.
- A flat irregular PED suspects (pachychoroid) neovascularization. Did authors find any association between treatment responses and the PED height or shape? In addition, please show the change of PED height or shape irradiated by SRT.
- Although the authors wrote “while no adverse events were observed after RFD-guided SRT, FFA was still useful in obtaining the overall range of appropriate preset pulse energy for SRT lesions” in the line 293, we don’t want to perform FFA just for the titration. Because the authors have treated so many cases, please indicate the range of laser power determined by FFA and RFD respectively and discuss any relationship between the laser settings and the fundus characteristics, or treatment responses.
- In the line 277, the authors wrote that the potency of the new RPE layer for removing SRF was confined to the SRT-treated area, and it would require more time to resolve a large SRF volume. If the SRT stimulates bioactive substances without any tissue damage as noted in the line 301, wider laser irradiation area, not only the leakage site, may accelerate earlier resolution of fluid. Please discuss how to manage the cases predicting poor responses for SRT.
Round 2
Reviewer 1 Report
Authors have addressed the critical points raised by the reviewer.
There are a few points to be addressed.
- Line 313: “Additionally, the closure of leaking points by new RPE cells might be helpful for healthy RPE cells to remove SRF. Since retreatment might induce RPE cell proliferation and stimulate greater release of cell mediators, such as matrix metalloproteinase-2 and pigment epithelium-derived factor, in SRT-treated areas than in single SRT, retreatment might be useful to remove persistent SRF. Furthermore, wider irradiation could be considered for cases with predicted poor response to SRT since it expands the SRT-treated area and eventually stimulates more release of the cell mediator”
-->Authors added some key words such as MMP2 and PEDF. However, it is still unknown, how these mediators contribute to the fluid resorption in CSC. These mediators, like MMP and PEDF, have been considered mainly as the related factors in the treatment of AMD. With respect to the healing mechanism of CSC, the improvement of RPE tight junction (outer BRB), water transport function of RPE cells, and also some (unknown) effect to the choroid, are assumed. Of course the possible contribution of the mediators mentioned above cannot also be excluded.
It is recommended to base the discussion here more on the known pathological features of CSC. Moreover, and very importantly, please cite the relevant literatures.
- Line 332: “Regarding test spot evaluation, RFD detected 93.9% of FFA-positive spots in this study. RFD missed some spots mainly due to faint FFA-positive spots, but no adverse events were observed after RFD-guided SRT. Therefore, RFD-guided SRT did not induce overtreatment in the current study.”
--> Even though no adverse event was observed, about 6% is not negligible. If these 6% of the false nevative spots were all very faint FFA positive spots, authors should state so more clearly. If it does not apply to not all spots, even a few percents, then there is still a risk of overtreatment. Therefore, authors are encouraged to discuss the possible measures to prevent overtreatment except RFD. The current statement is not convincing enough.
- Line 339: “Although SRT is considered a type of SMPL because of common characteristics, such as “invisibility during irradiation” and “photoreceptor-sparing,” SRT uses much shorter micropulses and different wavelengths. Moreover, unlike SMPL, which stimulates bioactive substances without any tissue damage, SRT damages RPE cells selectively and induces formation of a new RPE layer at the SRT lesion. Therefore, our results are not generalizable to other SMPLs.”
-->This part is still not sufficient. The part of the “wavelength” should be deleted. This is not the essential difference. Even if the same wavelength would be used to SMPL, SRT could be possible. The largest and most essential difference is the difference of the laser pulse length, which leads to the difference in the mechanism of action for RPE cell death. SMPL, typically with the pulse length of 100 to 300 µs, may lead to the temperature rise of RPE cells during irradiation, and if the temperature rises highly enough, then cells are coagulated. On the other hand, SRT utilizes very short pulse length of 1.7 µs, which leads to the temperature rise only at the melanosomes and cells are destroyed mechanically through the microbublle formation.
It is thus recommended to re-phrase this part, describing about the essential differences in 1) the pulse length and 2) mechanism of actions (thermal or mechanical).
Reviewer 2 Report
The authors revised their manuscript adequately.
Author Response
Thank you for your insightful suggestion.